# The resolution of face perception varies systematically across the visual field

**Anisa Y. Morsi** [1]*, **Valérie Goffaux** [2,3], **John A. Greenwood** [1]

**1** Experimental Psychology, University College London, London, United Kingdom, **2** Psychological Sciences Research Institute, UCLouvain, Ottignies-Louvain-la-Neuve, Belgium, **3** Institute of Neuroscience, UCLouvain, Ottignies-Louvain-la-Neuve, Belgium

* annie.morsi.18@ucl.ac.uk

## Abstract

Visual abilities tend to vary predictably across the visual field–for simple low-level stimuli, visibility is better along the horizontal vs. vertical meridian and in the lower vs. upper visual field. In contrast, face perception abilities have been reported to show either distinct or entirely idiosyncratic patterns of variation in peripheral vision, suggesting a dissociation between the spatial properties of low- and higher-level vision. To assess this link more clearly, we extended methods used in low-level vision to develop an acuity test for face perception, measuring the smallest size at which facial gender can be reliably judged in peripheral vision. In 3 experiments, we show the characteristic inversion effect, with better acuity for upright faces than inverted, demonstrating the engagement of high-level face-selective processes in peripheral vision. We also observe a clear advantage for gender acuity on the horizontal vs. vertical meridian and a smaller-but-consistent lower- vs. upper-field advantage. These visual field variations match those of low-level vision, indicating that higher-level face processing abilities either inherit or actively maintain the characteristic patterns of spatial selectivity found in early vision. The commonality of these spatial variations throughout the visual hierarchy means that the location of faces in our visual field systematically influences our perception of them.

## Introduction

Vision varies across the visual field. The recognition of simple low-level stimuli (ranging from lines to letters) varies systematically, becoming worse with increasing distance from fixation [1] and at specific angular locations around fixation [2]. In contrast, the perception of high-level stimuli such as faces has been found to vary across the visual field in a unique or idiosyncratic fashion [3–5]. These distinct variations are consistent with the view that faces are "special" in the visual system [6, 7], and suggest that variations in high-level processing may arise independently from those of low-level vision. Comparison of these low- and high-level visual abilities is difficult due to differences in the methodology used to measure them, however. Here, we aligned the methodology used to investigate anisotropies for low-level stimuli and faces by measuring the spatial resolution of face perception around the visual field.

Low-level properties such as visual acuity (spatial resolution) not only decline with eccentricity [1] but also vary by location with eccentricity held constant. Acuity is typically better along

**Data Availability Statement:** The data, code and materials for all 3 experiments are publicly accessible at osf.io/dxg96.

**Funding:** This work was supported by the Biotechnology and Biological Sciences Research Council [grant number BB/J014567/1].

**Competing interests:** The authors have declared that no competing interests exist.

the horizontal meridian vs. the vertical [8–10] and in the lower vs. the upper visual field [8, 9]. These horizontal-vertical and upper-lower anisotropies consistently emerge for many elements of vision, including orientation discrimination and contrast sensitivity [11–13], and have been linked with the retinotopic organisation of the visual system [14]. At the retinal level, the density of retinal ganglion cells is higher along the horizontal vs. vertical meridian [15, 16]. In early visual cortex (V1-V3), smaller population receptive fields (pRFs) have been found along the horizontal vs. vertical meridian and the lower vs. upper field, highlighting variations in visual field sampling [17, 18]. Higher cell densities and smaller pRF sizes have been associated with better acuity [19]. In this way, low-level vision is fundamentally influenced by location.

Variations in face recognition appear to differ substantially from those of low-level vision. Some studies report faster recognition of face gender in the upper vs. the lower field [5, 20]. Others report no significant horizontal-vertical or upper-lower differences in discriminating facial identity, instead observing an advantage for the left vs. right visual field [4], consistent with left hemifield biases in face perception [21, 22]. Finally, biases in the apparent gender, age, and identity of morphed faces have been found to vary across the visual field in an entirely idiosyncratic manner across participants [3, 23]. These distinct and/or idiosyncratic patterns suggest a dissociation in the mechanisms driving the visual field variations in low-level vision and face perception.

This dissociation may not be surprising given evidence that faces undergo distinct forms of processing [6]. Relative to other objects, face recognition is disproportionately impaired for upside-down vs. upright faces [24]. This *inversion effect* is driven by increased sensitivity to the spatial relationships between features (configural processing) within upright faces [7, 25, 26]. Neuroimaging has identified ventral occipitotemporal brain regions dedicated to face processing, including the fusiform face area (FFA) which shows greater activation for upright vs. inverted faces [27, 28]. These higher-level face-selective regions nonetheless show retinotopic sensitivity, with smaller receptive fields in the fovea vs. the periphery [29, 30]. Given the dissociations between low-level vision and face perception however, it is unclear how this selectivity is linked to that of earlier stages in the visual hierarchy.

A major challenge in comparing variations in low- and high-level vision derives from methodological differences. While measurements of low-level vision focus on spatial properties such as acuity, face recognition is usually measured via appearance-based judgements of fixed-size faces [3–5]. To align these approaches, we developed an acuity test for faces, measuring the smallest size necessary to judge gender at each visual field location. If face processing systems share the spatial properties of early visual cortex, anisotropies similar to those found for low-level vision should emerge for gender acuity. This pattern could arise either because face recognition systems inherit these variations from earlier stages or because face-selective brain regions actively maintain the same anisotropies. Alternatively, gender acuity could show either idiosyncratic [3, 23] or systematic patterns of variation that are unlike those of low-level vision [4, 5]. The latter outcomes would suggest that face perception involves distinct mechanisms that do not inherit the spatial selectivity of earlier brain regions. To determine the engagement of face-selective processes in our task, we measured gender acuity with both upright and inverted faces. If the task were solely limited by low-level acuity, the recognition of upright and inverted faces should not differ. Face-selective processes would instead be revealed by the characteristic advantage for upright faces [24].

# Experiment 1

## Method

**Participants.** 14 participants (13 female, 1 male, $M_{age}$ = 24.9 years) took part, including authors AYM and JAG; the rest were naïve. All had normal or corrected-to-normal foveal

vision of at least 20/20, assessed using a Snellen chart. 9 were right-eye dominant, determined using the Crider ring test [31]. This sample size was derived from previous studies with similar designs [11]. All three experiments were approved by the Research Ethics Committee for Experimental Psychology at University College London and all participants gave written informed consent before testing began.

**Apparatus.**    The experiment was programmed in MATLAB (MathWorks, Inc) and conducted on an Apple iMac running PsychToolbox [32–34]. Stimuli were viewed binocularly on a Cambridge Research Systems Display++ monitor with 2560 x 1440 resolution and 120 Hz refresh rate. The monitor was gamma corrected and linearised through software to have a minimum luminance of 0.16 cd/m$^2$ and a maximum of 143 cd/m$^2$. Participants were seated at a 50cm viewing distance, with head movements minimised using forehead and chin rests. The experiment took place in a dark room, and responses were recorded with a keypad.

**Stimuli.**    8 male and 8 female faces were selected from a bank of faces created by Laguesse and colleagues [35]. Because our task involved a binary gender judgement, we sought to avoid ambiguity by selecting faces at each end of the gender spectrum. This was ensured by taking faces that had received ratings above 8 out of 10 for either maleness or femaleness in a separate study (where faces were presented in a similar fashion to the current study–greyscale and within an oval aperture). All faces were grayscale, front-facing, and had a neutral expression (Fig 1B). Upright and inverted faces were included to determine whether face-specific configural mechanisms were engaged, and to assess whether visual field variations would differ with inversion. Using Adobe Photoshop CS6, each face was edited into an egg-shaped aperture measuring 657 x 877 pixels (at its widest and highest point, respectively) so that the only differences between images were due to internal features and not outer face shape (e.g. jawline). Pilot testing was conducted to ensure that gender-recognition performance was broadly

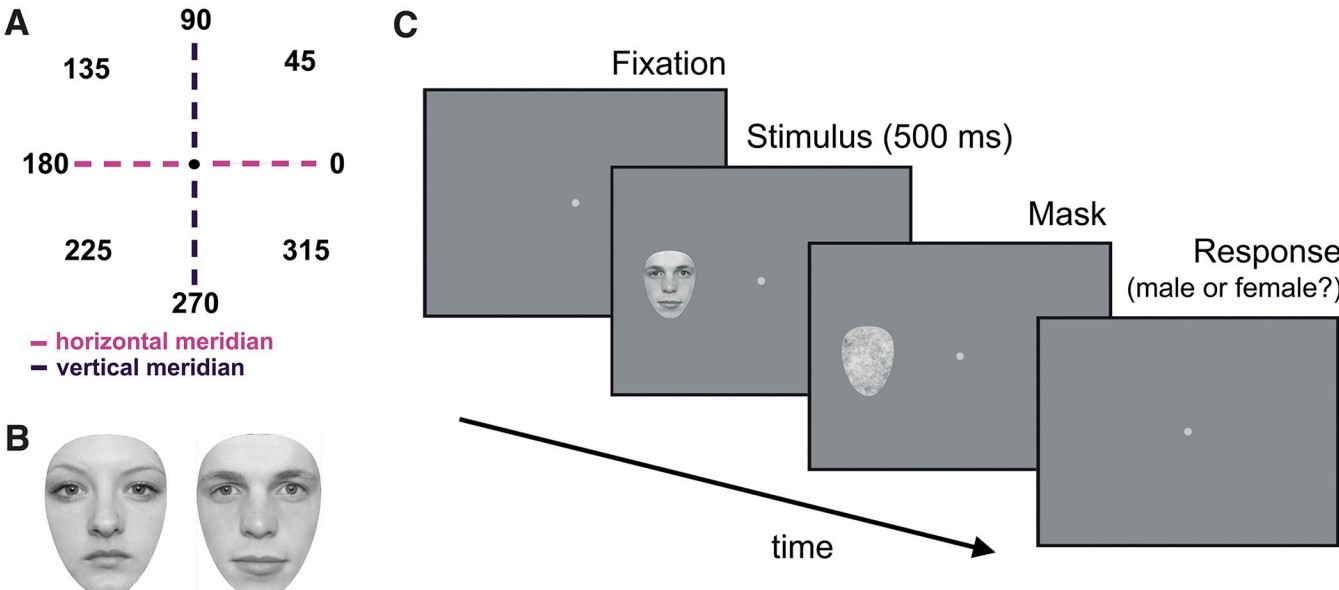

**Fig 1.** **(A)** The 8 polar angles tested, starting with 0˚ in the right visual field and preceding counterclockwise in 45˚ increments. The horizontal and vertical meridians are represented by pink and purple dashed lines, respectively. **(B)** Examples of female (left) and male (right) faces. **(C)** Experimental paradigm. A Gaussian fixation point first appeared, then a face was presented for 500 ms at one of the eight possible locations, selected randomly (shown here at 180˚). Each face was followed by a mask which remained on screen until a keyboard response was made. Faces varied in size from trial to trial according to an adaptive QUEST+ procedure.

similar across the face set (to minimise the possibility that distinctive features in individual faces could drive performance). The faces were set to have the same mean luminance as the monitor, with matched root-mean square (RMS) contrast values of 0.68. This ensured that overall luminance or contrast values could not be used as cues to gender.

**Procedure.** Participants were instructed to fixate on a white two-dimensional Gaussian element (standard deviation of 13.8 minutes of arc) in the centre of the screen. During each trial (Fig 1C), a face was presented for 500 ms, with the image centre located at 10° eccentricity and at one of 8 possible angles (Fig 1A). The face was immediately followed by a 1/f noise egg-shaped mask, which broadly matches the spatial frequency content of faces and natural scenes [36]. The size of the mask varied trial-by-trial to match the size of the face just shown, remaining on screen until participants made their response. We used a single interval 2-alternative forced choice (2AFC) response method, with participants reporting the face as either male or female using a numeric keypad. Audio feedback was provided after each response.

Before experimental trials began, participants completed a shorter set of 72 practice trials to become accustomed to the task. For the practice trials, faces were presented at fixed sizes of 600, 400 and 200 pixels (face size refers to vertical height, with width scaled proportionately), with 9 trials at each location. Participants were required to be at least 90% correct in order to continue. During the experimental trials an adaptive QUEST procedure [37] varied face sizes presented at each location according to the participant's responses, set to converge on the size at which 75% of responses were correct. Within each block of trials, QUEST estimates were computed separately for each location. Faces were presented at sizes within ±1/3 of the QUEST threshold estimate on each trial (minimum 5 and maximum 640 pixels). This "jitter" allowed us to collect data for a range of sizes, which improved the subsequent fitting of psychometric functions to the data [38].

Each experimental block contained 50 faces shown at each of the 8 locations (with independent QUEST procedures) to give 400 trials in total. Each face was shown an equal number of times, in a randomised order, with the location it appeared at also randomised. Upright and inverted faces were presented in alternate blocks. The experiment consisted of 1–2 practice blocks, followed by 8 experimental blocks (4 repeats for both upright and inverted faces) to give 3200 trials in total. During analysis, we fit psychometric functions to the combined data from these 4 repeats (separately for each location and inversion condition).

**Analyses.** Responses were first sorted by face size (in pixels) and collated in 20-pixel bins (e.g. faces of 8, 15 and 18 pixels would fall in the same bin). The proportion of correct responses was then calculated for each face-size bin. Cumulative Gaussian functions (Fig 2) were fit to these data using 3 free parameters for the mean, variance and lapse rate [39]. The lapse rate was set to a maximum of 0 by default. For some participants whose responses did not reach ceiling at the largest face sizes, the maximum allowable lapse rate was increased first to 0.05 and then 0.1 in order to improve curve fitting (required for 7 participants in Experiment 1, 3 in Experiment 2, and none in Experiment 3). Importantly, this was applied on an individual basis (equally across all conditions), meaning that within-subjects variations across locations or conditions cannot be attributed to this factor. Gender acuity thresholds for each location were taken as the size at which 75% accuracy was reached, then converted from pixels to degrees of visual angle.

Statistical analyses were carried out using a 3-way mixed effects analysis of variance (ANOVA), with participant as a between-subjects random factor and inversion (upright, inverted) and location (0, 35, 90, 135, 180, 225, 270, 315°) as within-subjects fixed factors. A priori comparisons took the form of repeated-measures t-tests, comparing thresholds between the horizontal (0 and 180°) vs. vertical (90 and 270°) meridians, the upper (90°) and lower (270°) field, and the left (180°) and right (0°) locations, for upright and inverted faces separately.

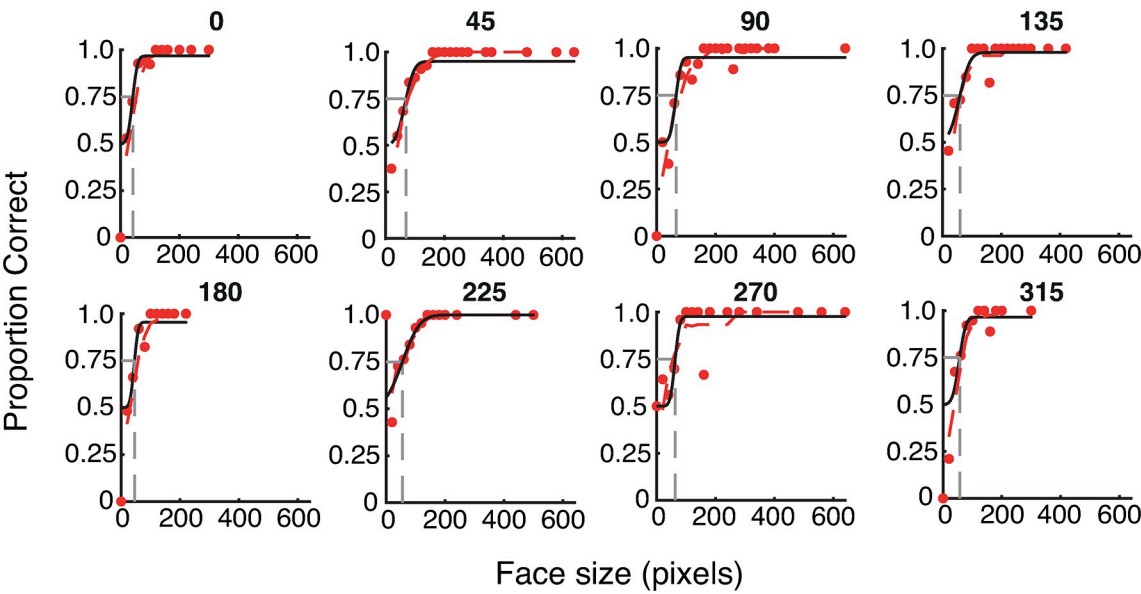

**Fig 2. Psychometric functions for a single participant, showing the proportion of correct gender judgements for different sized upright faces at each of the 8 visual field locations (labelled at the top of each graph, where 0 = rightwards, 90 = upwards, etc.).** Performance improves monotonically as a function of face size. Dashed grey lines plot thresholds for gender acuity (the size at which 75% accuracy was reached).

## Results

Mean gender acuity thresholds across participants are plotted as both a bar chart and according to the polar angle of each of the 8 locations in Fig 3. Smaller values represent better gender acuity. Mean gender acuity thresholds were worse for inverted compared to upright faces overall, indicating that inversion disrupted the ability to judge gender at all locations. There was a sizeable difference in gender acuity according to location; for upright faces, there was a range of almost 2° of visual angle between the smallest threshold value in the lower field (270°) and the largest in the upper field (90°). Thresholds were smaller along the horizontal (0, 180°) as opposed to the vertical (90, 270°) meridian. Thresholds at the diagonal locations varied inconsistently, with smaller thresholds in the top left (135°) vs. top right (45°) location, but larger thresholds in the bottom left (225°) vs. bottom right (315°) location.

The ANOVA revealed a main effect of location, $F(7,91) = 3.41$, $p = .003$, $d = 0.21$, confirming that the location of faces in the visual field influenced gender perception. Planned contrasts revealed that thresholds were significantly smaller along the horizontal (0° and 180° averaged) compared to the vertical (90° and 270° averaged) meridian for both upright, $t(13) = -2.84$, $p = .014$, and inverted faces, $t(13) = -2.21$, $p = .046$. Thresholds were also smaller in the lower compared to the upper field for upright faces, $t(13) = 2.68$, $p = .019$, although not for inverted faces, $t(13) = -0.10$, $p = .923$. There was no difference between thresholds at the left and right locations for upright, $t(13) = -1.79$, $p = .096$, or inverted faces, $t(13) = -0.61$, $p = .551$. In other words, we observe both horizontal-vertical and upper-lower anisotropies for gender acuity, though performance did not differ between left and right hemifields.

The error bars in Fig 3 indicate that there was considerable between-participants variability in gender acuity thresholds, with the ANOVA showing a main effect of participant, $F(7,91) = 2.98$, $p = .029$, $d = 0.75$. However, there was no interaction between location and participant, $F(91,91) = 1.07$, $p = .383$, $d = 0.52$, indicating that individuals varied in their overall threshold

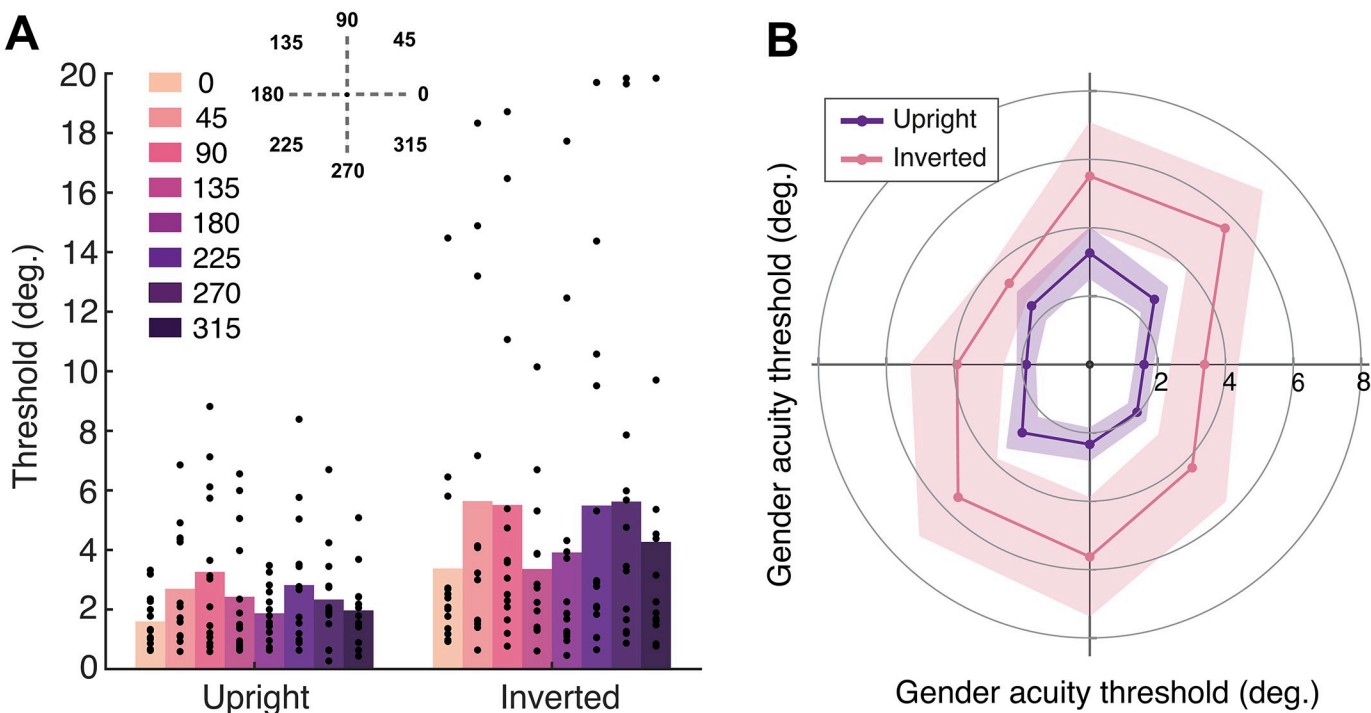

**Fig 3.** Mean gender acuity thresholds (in degrees of visual angle) measured in Experiment 1, plotted in two ways. Firstly, as a bar graph **(A)** with each angular location indicated via colour (see legend). Individual data points represent thresholds for each participant. Secondly, as a polar angle plot **(B)**, where 0˚ is at the right and angles proceed counterclockwise in 45˚ jumps. Upright faces are shown in purple and inverted in pink. Shaded regions denote ± 1 SEM.

magnitude rather than exhibiting idiosyncratic variations across the 8 locations. We note in particular that a subset of participants showed thresholds that were considerably smaller (with values ~0.5˚) than the rest of the group.

The presence of an inversion effect was supported by a main effect of inversion, $F(1,13) = 6.61$, $p = .023$, $d = 0.34$, showing that thresholds were significantly higher for inverted compared to upright faces. Therefore, the processing of configural information in upright faces appears to have benefitted performance in our task. There was no interaction between inversion and location, $F(7,91) = 0.91$, $p = .506$, $d = 0.07$, indicating that inversion disrupted gender perception to a similar extent across the visual field. There was, however, a significant interaction between inversion and participant, $F(13,91) = 10.55$, $p = < .001$, $d = 0.60$. To investigate this interaction further, we calculated mean face inversion effect (FIE) values across participants by subtracting upright from inverted thresholds. Upon closer analysis the inversion-participant interaction appeared to be driven by two participants with very large FIEs, and indeed we found that removing their data from the analysis eliminated the interaction.

## Experiment 2

Experiment 1 demonstrates that face recognition acuity varies in the same way as low-level vision. We had two concerns, however. First, the low thresholds of some participants (as small as 0.5˚) suggests they may have occasionally fixated the faces. Second, the eyes of our upright face stimuli were closer to fixation in the lower vs. upper field, which could have driven the upper-lower difference, given the importance of the eyes in gender perception [40, 41]. In

Experiment 2 we sought to validate our findings by adding eye-tracking and fixing eye position across locations.

## Method

**Participants.** 14 participants (12 female, 2 male, $M_{age}$ = 23.6 years) took part, including author AYM; the rest were naïve and newly recruited. All had normal or corrected-to-normal vision. 7 were right-eye dominant.

**Apparatus.** In addition to the setup in Experiment 1, we used an EyeLink 1000 (SR Research, Mississauga, ON, Canada) to monitor fixation during trials.

**Stimuli.** The same face stimuli from Experiment 1 were used. Stimulus locations were centred on the eyes themselves, so that the centre of the eyes was always 10° from fixation regardless of face size, angular location or inversion (Fig 4). The position of each face in the

## Experiment 1

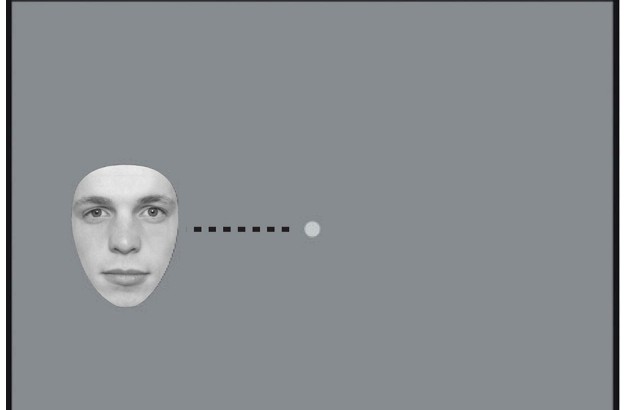

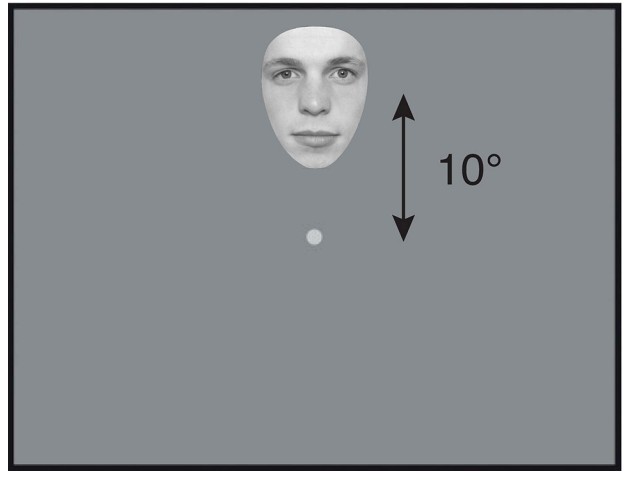

## Experiment 2

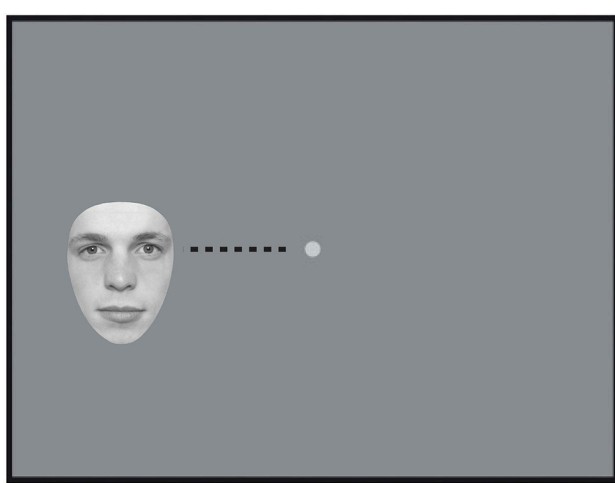

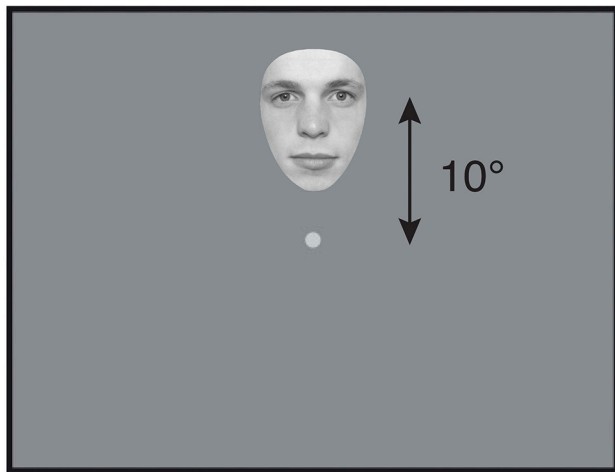

**Fig 4.** In Experiment 1 (left panels), faces were presented 10° from fixation according to the centre of the face. In Experiment 2 (right panels), the centre of the eyes was always 10° from fixation regardless of face size, angular location or inversion.

egg aperture was also shifted and/or rotated slightly to ensure that the eyes fell at the same point within the aperture.

**Procedure.** Following calibration of the EyeLink to track their left eye, participants were required to fixate the Gaussian element (with an allowable error of 1.5° radius) in order for each trial to start. Trials in which fixation diverged from this region were cancelled and repeated at the end of the block. Participants first completed the practice block(s), as in Experiment 1. Experimental blocks were split according to whether faces were upright or inverted and locations were cardinal (0, 90, 180, 270°) or diagonal (45, 135, 225, 315°) angles, resulting in 4 blocks: upright cardinal, upright diagonal, inverted cardinal and inverted diagonal. These split blocks were introduced so that the blocks would not be too long, as the eye tracking increased the duration of data collection depending on the amount of cancelled trials and recalibration needed. Data were collected over 4 hour-long sessions, with each of the 4 conditions repeated once per session. This gave a total of 16 blocks and 4096 trials per participant.

## Results

Mean gender acuity thresholds are plotted in Fig 5. Again, smaller values represent better gender acuity. Compared to Experiment 1, thresholds in Experiment 2 were higher overall and had reduced variability across participants (particularly in the inverted condition), suggesting that eye tracking successfully stopped participants from looking directly at faces. Indeed, the smallest-measured threshold in Experiment 2 was 1.31°, compared with 0.27° in Experiment 1.

The ANOVA revealed a main effect of location, $F_{(7,91)} = 5.55$, $p = < .001$, $d = 0.30$, indicating that gender acuity was influenced by the location of faces. There was a clear horizontal-vertical difference, with planned comparisons revealing that thresholds were significantly smaller along the horizontal (0° and 180° averaged) compared to the vertical (90° and 270° averaged) meridian, for both upright, $t(13) = -6.16$, $p < .001$, and inverted faces, $t(13) = -3.00$, $p = .010$. However, although thresholds were smaller in the lower compared to the upper field, the difference was not significant for either upright, $t(13) = 1.19$, $p = .256$, or inverted faces, $t(13) = 0.38$, $p = .713$. Similarly, thresholds did not differ between the left and right locations for either upright, $t(13) = -0.10$, $p = .926$, or inverted faces, $t(13) = 0.39$, $p = .704$.

The engagement of face-selective processes can again be seen with the inversion effect in the mean data (Fig 5A and 5B), with higher thresholds and therefore reduced ability to perceive gender (i.e. larger faces needed) for inverted compared to upright faces overall. This was confirmed by a significant main effect of inversion, $F_{(1,13)} = 17.93$, $p = .001$, $d = 0.58$. There was no interaction between inversion and location, $F_{(7,91)} = 1.29$, $p = .265$, $d = 0.09$, indicating that inversion effects did not differ significantly across the visual field.

Gender recognition abilities again differed between individuals, which was confirmed by a main effect of participant, $F_{(7,91)} = 11.52$, $p < .001$, $d = 0.93$. There was however no interaction between location and participant, $F_{(91,91)} = 0.94$, $p = .623$, $d = 0.48$, indicating that location-based variations in gender perception are not specific to the individual. In other words, face perception differed across the visual field in a characteristic pattern shared across individuals.

Unlike the previous experiment there was no interaction between inversion and participant, $F_{(13,91)} = 1.60$, $p = .100$, $d = 0.19$, indicating that individuals did not vary substantially in the inversion effect. This suggests that the significant interaction in Experiment 1 may have been caused by a subset of participants looking at the faces–accordingly, these individuals had low thresholds for both upright and inverted faces (i.e. little to no inversion effect), likely driven by their fixating the faces in both conditions.

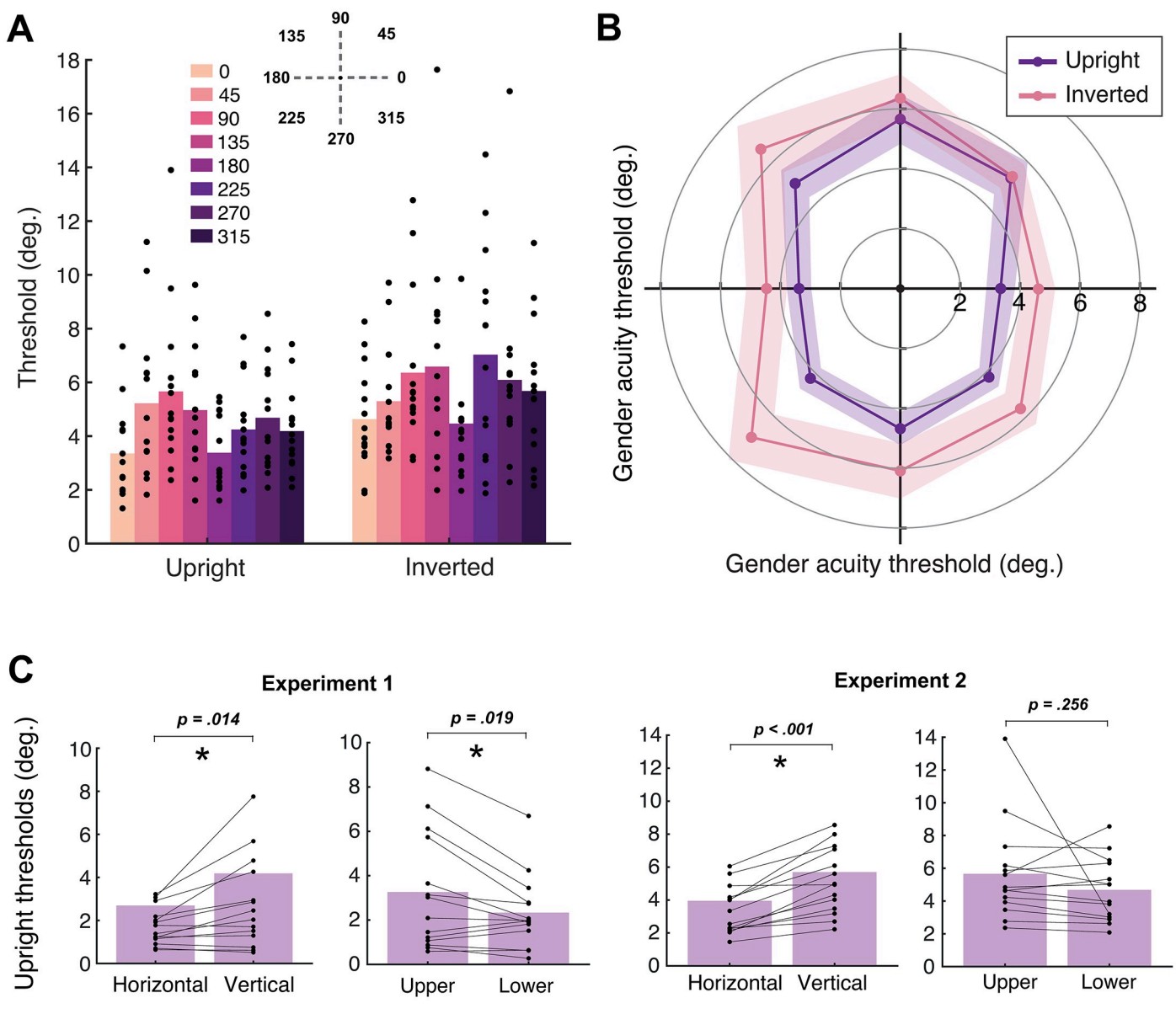

**Fig 5.** Mean gender acuity thresholds from Experiment 2, first shown as a bar graph **(A)** with each location indicated via colour (see legend). Individual data points represent thresholds for each participant. Mean thresholds are also visualised in a polar angle plot **(B)**, with 0° at the right and angles increasing counterclockwise by 45° each time. Upright faces are shown in purple and inverted in pink. Shaded regions represent ± 1 SEM. **(C)** Bar charts comparing the horizontal-vertical difference and upper-lower difference in Experiments 1 and 2. Data are plotted for upright faces only. Horizontal refers to thresholds averaged across 0° and 180° locations, with vertical the average of 90° and 270°. Upper represents 90° and lower 270°. Individual data points represent thresholds for each participant. Significant differences ($p < .05$) are marked with an asterisk.

To compare anisotropies more clearly, bar charts displaying the horizontal-vertical difference and upper-lower difference in both experiments are shown in Fig 5C. Only data for upright faces are included. The charts on the left-hand side show that the horizontal-vertical difference was consistent across both experiments, with significantly lower thresholds (better gender acuity) for faces at horizontal compared to vertical locations. The upper-lower

difference was only significant in Experiment 1, where effects could have been driven by eye movements or variations in eye position within face stimuli. Although a trend in the same direction persisted when we controlled for these factors in Experiment 2 –showing that the upper-lower difference cannot be attributed to these factors alone–the difference was no longer significant.

To summarise, gender acuity was better along the horizontal vs. vertical meridian but did not differ significantly in the lower vs. upper field. However, even in Experiment 1 the upper-lower difference was smaller than the horizontal-vertical difference, suggesting that it may simply be harder to measure. We examined this possibility in Experiment 3.

## Experiment 3

Given the trend towards better gender acuity in the lower vs. upper field in Experiment 2, we conducted further measurements at these locations to determine whether a significant difference would emerge with a greater number of trials.

### Method

**Participants.** 14 participants (11 female, 2 male, 1 non-binary, $M_{age}$ = 22.1 years) took part, including author AYM; the rest were naïve and newly recruited. All had normal or corrected-to-normal vision. 7 were right-eye dominant.

**Stimuli.** Stimuli were as in Experiment 2, with faces shown at the upper (90˚) and lower (270˚) locations only.

**Procedure.** Blocks were split according to whether faces were upright or inverted, with 128 trials in each block. Data were collected over 2 hour-long testing sessions, with each of the 2 conditions (upright/inverted) repeated 8 times per session (with an extra block completed at the start of the first session, which acted as a practice block and was not included in data analysis). This gave a total of 16 experimental blocks and 2048 trials over the experiment (doubling the number of trials per location compared to Experiment 2). Remaining parameters were as in Experiment 2.

### Results

Mean gender acuity thresholds are plotted in Fig 6, with smaller values representing better gender acuity. The ANOVA revealed a main effect of location, $F(1,13) = 22.97$, $p < .001$, $d = 0.91$, indicating that gender acuity differed between the upper and lower fields. Gender acuity thresholds were significantly smaller in the lower field compared to the upper for upright faces, $t(13) = 3.82$, $p = .002$, and approached significance for inverted faces, $t(13) = 2.07$, $p = .059$. On an individual level, 12 of 14 participants showed better acuity in the lower vs. the upper field. These results highlight the presence of an upper-lower difference in face recognition, with better gender acuity in the lower half of the visual field.

Like the previous two experiments, there was a main effect of orientation, $F(1,13) = 8.38$, $p = .013$, $d = 0.39$, indicating an overall inversion effect whereby gender acuity thresholds were larger for inverted compared to upright faces, $t(27) = -2.92$, $p = .007$. There was also a main effect of participant, $F(1,13) = 31.12$, $p = .032$, $d = 1.00$, again highlighting overall differences in gender acuity between individuals. There was no significant interaction between location and participant, $F(1,13) = 0.58$, $p = .829$, $d = 0.37$, suggesting that there was a common pattern of gender acuity across individuals. Interactions were similarly non-significant for location and orientation, $F(1,13) = 0.91$, $p = .358$, $d = 0.07$, and orientation and participant, $F(1,13) = 1.02$, $p = .484$, $d = 0.51$, indicating that gender acuity patterns were similar for upright and inverted faces, and that inversion effects did not vary significantly between individuals.

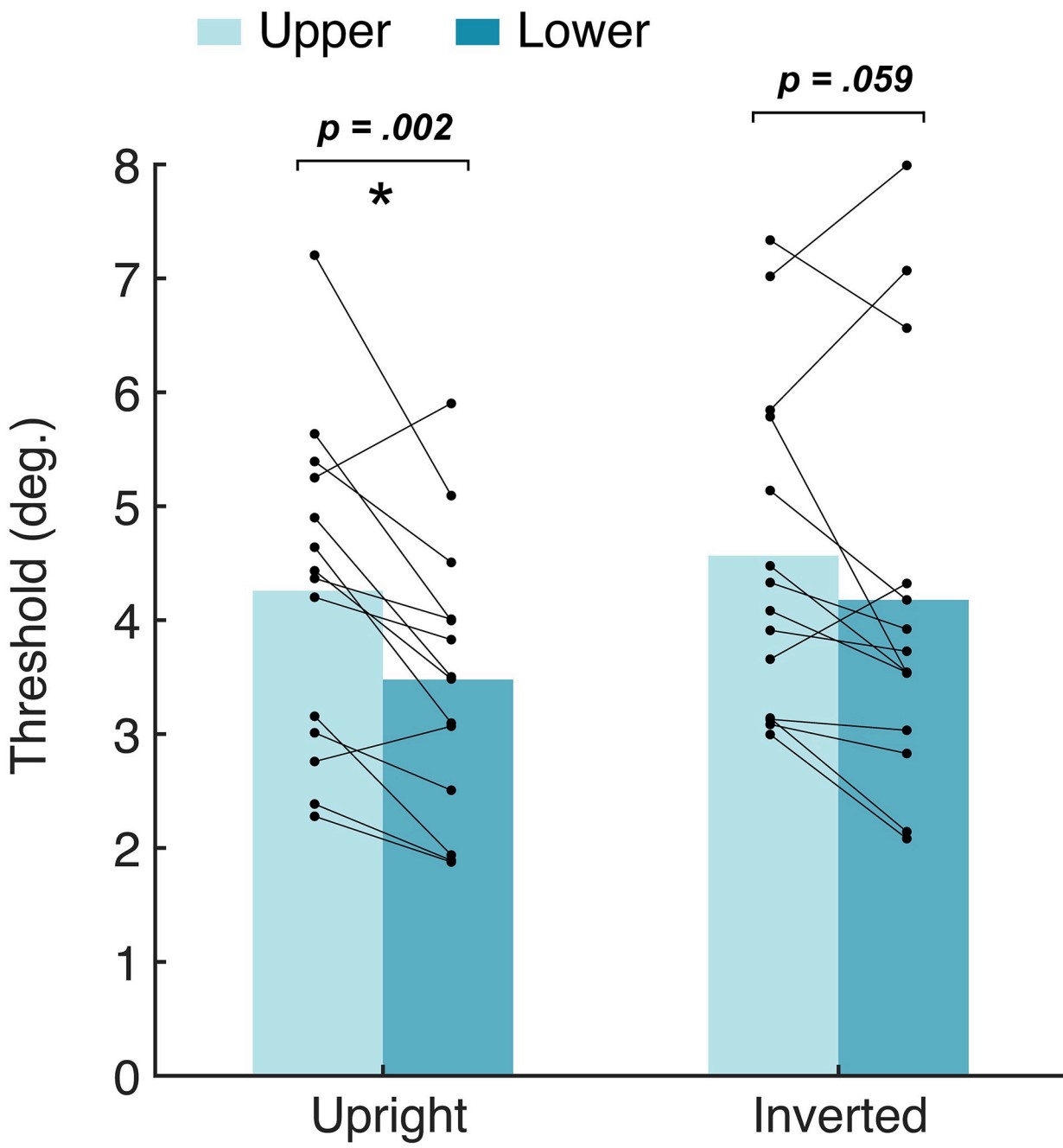

**Fig 6. Mean gender acuity thresholds for the upper (90˚; lighter blue) versus lower (270˚; darker blue) visual field, for both upright and inverted faces.** The asterisk represents a significant difference ($p < .05$). Individual data points represent thresholds for each participant, with lines connecting their performance between upper and lower locations.

## Discussion

We demonstrate that face perception varies across the visual field in a systematic pattern. Across 3 experiments, acuity for gender judgements showed a horizontal-vertical anisotropy for both upright and inverted faces, where recognition was possible with smaller faces on the

horizontal vs. vertical meridian, and a small-but-reliable upper-lower difference, with better acuity in the lower vs. upper field. The presence of both these anisotropies, and the smaller magnitude of the upper-lower difference, matches the patterns of low-level vision [8, 11, 12, 42] and demonstrates that the resolution of face perception varies predictably across the visual field, rather than uniquely or idiosyncratically [3–5]. This suggests that spatial properties are preserved throughout the visual hierarchy, including in higher-level face-selective systems.

Our observation of systematic variations in face recognition differs from prior studies reporting purely idiosyncratic variations around the visual field. Prior studies demonstrating these idiosyncrasies have relied on judgements of *appearance* [3, 23]. Judgements of apparent size [43] and position [44] have similarly revealed perceptual idiosyncrasies in low-level vision, likely related to localised visual-field distortions. These individual-dependent distortions could in fact drive idiosyncrasies in facial-identity judgements (e.g. by altering facial features), given that the idiosyncrasies for one set of faces do not correlate with those for other faces [23]. The use of a single male/female pair to measure gender judgements in these prior studies [3] would be similarly susceptible to individual-based distortions. In contrast, our measurement of *performance* with multiple unambiguously gendered faces is less susceptible to smaller individual-dependent variations, allowing us to uncover larger systematic anisotropies.

Our results also differ from the systematic-but-unique variations found for faces previously. For instance, judgements of the identity of synthetic-contour faces show a left visual-field advantage [4], whereas we found no difference between the left and right fields. Our finding is nonetheless consistent with the broader observation that face perception is not reliably lateralised, but rather that these effects depend on the task [45–48] and/or stimuli, with the clearest effects emerging for large, chimeric faces that span fixation [22, 49]. Additionally, whereas we observe better gender acuity in the lower vs. upper field, prior studies report faster gender recognition in the upper field [5, 20]. This difference likely reflects our measurement of performance rather than reaction times, given that an upper-field advantage is also evident in the temporal response to low-level stimuli [50, 51]. Altogether, our findings show that when the spatial resolution of face perception is measured in the same way as lower-level visual abilities, similar visual field anisotropies emerge.

In Experiments 2 and 3, the upper-lower difference was reduced (relative to Experiment 1) when the position of the eyes was matched across locations, confirming a particular importance of the eyes for gender perception [40, 41]. Across all experiments, acuity thresholds were nonetheless consistently lower for upright vs. inverted faces. This inversion effect confirms that our gender acuity task sufficiently engaged face-specific mechanisms, consistent with prior reports that the configural processing of upright faces occurs across both foveal and peripheral vision [46, 47, 52]. Interestingly, while inversion effects are typically measured using same-size faces [24], here we highlight a spatial component to face processing [30, 60]. This ability to identify upright faces at smaller sizes than inverted faces could reflect the added benefit of configural processing with upright faces [6]. Importantly, this inversion effect indicates that the thresholds for gender acuity that we measure cannot be solely attributed to lower-level limitations on spatial vision such as visual acuity and contrast sensitivity, nor to the recognition of individual facial features such as the eyes.

Nevertheless, the anisotropies that we observe for face perception could be partly driven by variations in *featural selectivity*, including properties like the radial bias, whereby peripheral contrast sensitivity and orientation discrimination are better for stimuli oriented towards the fovea [53, 54]. Because horizontal information is particularly informative for face recognition [55–57], this could improve performance on the horizontal vs. vertical meridian–either passively through the pooling of variations in low-level information, or more actively by boosting the response to the optimal orientations for high-level processing, as argued recently [58].

However, though these factors could contribute to the horizontal-vertical anisotropy, these radial variations are matched across the upper and lower fields, making them unable to explain the upper-lower anisotropy.

The systematic anisotropies we observe for face recognition could also arise from variations in *spatial selectivity* within face-selective systems. In low-level vision, better acuity has been linked to smaller receptive fields and increased cortical magnification [17–19, 59]. This could drive improved acuity for faces because higher levels passively inherit enhanced low-level input from locations on the horizontal (vs. vertical) meridian, and in the lower (vs. upper) field. Unlike low-level vision, however, better face perception has been linked with *larger* pRF sizes and the resulting increase in visual field coverage within face-selective regions [30, 60]. The observed anisotropies could thus derive instead from the way that face-selective neurons actively sample the visual field. Either way, the common pattern of anisotropies in low- and high-level vision suggests that retinotopic sensitivity within face-selective regions is not entirely distinct from that of earlier brain areas.

Given these possibilities for both the passive inheritance and active maintenance of featural and/or spatial selectivity, to what extent are face processing systems themselves contributing to the observed anisotropies? Note first that the presence of a consistent inversion effect indicates that performance on our gender acuity task derives from processing within face-selective systems (at least in part), rather than stemming purely from lower-level limitations on vision. However, the size of this inversion effect did not vary significantly around the visual field, as one might expect if face-selective processes were themselves to vary, and contrary to the recent finding of larger inversion effects in an identity-recognition task on the horizontal meridian than the vertical [58]. It could be that while the strength of face-selective processing varies around the visual field (leading to variable inversion effects with fixed-size stimuli), the resolution of these face-selective processes does not. We nonetheless observed similar anisotropies for *both* upright and inverted faces. These acuity thresholds are many times larger than those for the identification of simple elements–compared to data from Anstis [61], thresholds in Experiment 2 were approximately 8–13 times larger for upright faces, and 10–16 times larger for inverted faces. We suggest that face processing systems sample large regions of the visual field in order to support gender recognition, with greater efficiency for upright faces than inverted (and a requirement for broader sampling of information with inverted faces, given that inverted faces required larger stimulus sizes to reach threshold) and variations in sampling for both processes around the visual field. The spatial pooling required to extract holistic and/or configural information does not however appear to vary across the visual field.

In conclusion, face perception varies around the visual field with both horizontal-vertical and upper-lower anisotropies, matching patterns consistently found for low-level vision [8, 11, 12], and contrary to suggestions that face recognition varies in a unique or idiosyncratic manner [3–5]. Our results are consistent with a hierarchical model of face processing whereby the selectivity for faces is built on the selectivity of earlier levels. The variations in face perception that we observe are likely driven by variations in spatial selectivity, and perhaps in part by variations in featural selectivity, which may be inherited passively from earlier stages and/or actively maintained in face-selective regions. Ultimately, we demonstrate that common spatial variations are found throughout the visual system, causing location to systematically influence face perception.

## Acknowledgments

We thank Alexia Roux-Sibilon at UCLouvain for supplying us with the face images.

## Author Contributions

**Conceptualization:** Anisa Y. Morsi, Valérie Goffaux, John A. Greenwood.

**Data curation:** Anisa Y. Morsi.

**Formal analysis:** Anisa Y. Morsi, John A. Greenwood.

**Funding acquisition:** John A. Greenwood.

**Investigation:** Anisa Y. Morsi, John A. Greenwood.

**Methodology:** Anisa Y. Morsi, Valérie Goffaux, John A. Greenwood.

**Project administration:** Anisa Y. Morsi.

**Resources:** John A. Greenwood.

**Software:** John A. Greenwood.

**Supervision:** John A. Greenwood.

**Visualization:** Anisa Y. Morsi, John A. Greenwood.

**Writing – original draft:** Anisa Y. Morsi.

**Writing – review & editing:** Anisa Y. Morsi, Valérie Goffaux, John A. Greenwood.

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
