## [Decision Letter · Decision Letter 0]

17 Jan 2024

PONE-D-23-36530The resolution of face perception varies systematically across the visual fieldPLOS ONE

Dear Dr. Morsi,

Thank you for submitting your manuscript to PLOS ONE. After careful consideration, we feel that it has merit but does not fully meet PLOS ONE’s publication criteria as it currently stands. Therefore, we invite you to submit a revised version of the manuscript that addresses the points raised during the review process.

We look forward to receiving your revised manuscript.

Kind regards,

PremNandhini Satgunam

Academic Editor

PLOS ONE

Journal Requirements:

"This work was supported by the Biotechnology and Biological Sciences Research Council [grant number BB/J014567/1]."

"We thank Alexia Roux-Sibilon at UCLouvain for supplying us with the face images. This work was supported by the Biotechnology and Biological Sciences Research Council [grant number BB/J014567/1]."

"This work was supported by the Biotechnology and Biological Sciences Research Council [grant number BB/J014567/1]."

Additional Editor Comments:

**ACADEMIC EDITOR:**

Authors should carefully consider the comments of Reviewer 1. Especially on some important concerns raised with the methodology. I hope these concerns can be satisfactorily addressed in your revision.

Reviewers' comments:

Reviewer's Responses to Questions

**Comments to the Author**

1. Is the manuscript technically sound, and do the data support the conclusions?

Reviewer #1: Partly

2. Has the statistical analysis been performed appropriately and rigorously? 

Reviewer #1: Yes

3. Have the authors made all data underlying the findings in their manuscript fully available?

Reviewer #1: Yes

4. Is the manuscript presented in an intelligible fashion and written in standard English?

Reviewer #1: Yes

5. Review Comments to the Author

Reviewer #1: Morsi et al examine the visual field dependence of gender discrimination in faces. Performance on many visual tasks is strongly dependent on visual field location; in addition to a general loss in spatial resolution as eccentricity increases, asymmetries have been well-documented in horizontal/vertical and upper/lower visual fields across many tasks. For face perception, there have been inconsistent findings of visual field dependence.

Morsi et al. develop a face acuity task, in which threshold face size is estimated in a 2AFC gender discrimination task for standardized faces, presented upright or inverted at 10 deg eccentricity. Systematic visual field biases were found in Experiment 1: face size thresholds were smaller in left & right than upper & lower visual field, and were smaller in lower than upper visual field. These results were broadly replicated in Experiment 2, with the important addition of eye tracking to ensure compliant fixation and adjustment for eye eccentricity, and in Experiment 3, in which statistical power was increased by concentrating trials in the upper and lower visual field only. Across conditions, there was a strong face inversion effect, in which threshold size was lower for upright than inverted faces, with the same pattern of results as upright faces.The authors conclude that face processing varies across the visual field, but the effect is inherited from low-level sensitivity changes.

This is a relatively straightforward investigation of a robust effect that extends our understanding of visual field dependence for a more complex task than optotype acuity or contrast sensitivity. The experiments are mostly well-designed (once Experiment 2 corrected an issue in Experiment 1), the statistical analyses are appropriate and the manuscript is clearly written. However there are a number of methodological problems that weaken the interpretation of the results.

The conclusion of the paper seems to be that face perception per se does not vary systematically across the visual field - performance only changes because of low-level differences in contrast sensitivity.This conclusion is not well encapsulated by the current title.

The whole study is based on only 8 male and 8 female faces that were viewed at different sizes for many hundreds of trials by each participant. There is a high probability that participants could have learned the faces based on a small number of conspicuous features and based their response on those. This is especially problematic given that there was audio feedback for correct/incorrect responses. If so, the visibility of those features would depend on contrast sensitivity rather than facial configuration, and this would explain the results.

The studies were repeated at a single eccentricity (10 degrees). Letter acuity at 10 degrees is approximately 0.43 degrees (Anstis, 1974), so face thresholds are around 6-8 times larger than letter acuities. It would be beneficial for the author’s argument to report if face size acuity scales as a constant ratio of letter acuity with eccentricit .

Face gender was determined in a preliminary study to ensure >8/10 ratings. Were those trials completed with oval-windowed grayscale faces?

Line 170 The maximum allowable lapse rate of either 0.05 or 0.1 was applied to improve curve fitting. How was the improvement in curve fitting defined and what criterion was selected for an acceptable fit?

6. PLOS authors have the option to publish the peer review history of their article (what does this mean?). If published, this will include your full peer review and any attached files.

Reviewer #1: No

---

## [Author Response · Author response to Decision Letter 0]

27 Mar 2024

Firstly, we wish to thank the editor and reviewer for their comments on our manuscript. We have made a number of changes to improve both the clarity in the description of our methods, and the explanations of what our results may suggest. Below, reviewer comments are in blue and our responses are in black. In the revised manuscript, new sections have been highlighted with tracked changes and comments.

Our aim with the title is to refer to face perception more generally, broadly emphasising that our ability to resolve faces varies around the visual field systematically. This refers to our finding that acuity thresholds for both upright and inverted faces show the characteristic horizontal-vertical and upper-lower visual field anisotropies, and contrasts this result with the prediction from prior work that face perception might vary with e.g. completely idiosyncratic patterns across individuals. Our findings then naturally raise the question of the origin of these variations. While we agree that variations in low-level factors may indeed contribute to the anisotropies measured for faces (e.g. the radial bias, as discussed on p16, lines 402-412), the presence of large inversion effects demonstrates that face-specific processes were engaged by our gender recognition task. If performance on our task were solely due to low-level limitations such as contrast sensitivity, we would not observe such a strong inversion effect. 

Although the size of this inversion effect did not vary significantly around the visual field, as one might expect if face-selective processes were themselves to vary, we nonetheless observed anisotropies for the recognition of both upright and inverted faces. In other words, face-recognition performance varies around the visual field (as referred to in the title), but it is less clear whether the ‘special’ configural/holistic aspects of face perception vary in addition to this (at least as they contribute to the resolution of face perception). We interpret this pattern of results as suggesting firstly that face perception requires sampling across large regions of the visual field (larger than e.g. letter acuity, as the reviewer notes below). This sampling has greater efficiency for upright faces than inverted (given that inverted faces required larger stimulus sizes to achieve the same accuracy level), with variations in sampling for both processes around the visual field. However, the spatial pooling required to extract holistic and/or configural information does not however appear to vary across the visual field in addition to this baseline pattern of variations.

Relevant to the reviewer’s comment on acuity below, a similar argument can be made for variations in low-level visual acuity, which are limited by retinal (photoreceptor density) as well as neural factors (V1 receptive field size). We similarly suggest that face acuity likely involves limits on resolution within lower (e.g. the radial bias) as well as later stages of visual processing (e.g. neural properties of face-selective regions), which both contribute to variations in face perception. As additional work is needed to ascertain the exact origin of these anisotropies, we would prefer to leave the title as it is – referring to the variations in thresholds that we observe. To clarify these points, we have more strongly emphasised the importance of the observed inversion effects in this regard, both in the introduction to set up the experiments (pp4-5, lines 104-110), in the results (e.g. Experiment 1: p9, lines 229-232) and in the discussion (pp15-16). We have also clarified the overall interpretation of our results and the potential low- and high-level factors that may contribute to this with an additional paragraph in the discussion (pp16-17, lines 437-458). 

We were also concerned that the use of a relatively small number of faces for the gender judgement could have introduced biases towards individual faces and/or features. To minimise the presence of any particularly distinctive features within our face sets (both at the lowest level in terms of local variations in contrast and in terms of distinctive facial features like noses etc), we ran pilot tests to ensure that all faces were recognised with broadly similar accuracy levels (p6, lines 133-135). More importantly, if participants categorised face gender based solely on certain conspicuous face features, we should not see inversion effects. The fact that our 3 experiments consistently produced clear inversion effects suggests that we were indeed able to engage the use of face-selective processes with our paradigm, and that these judgements certainly cannot be solely reduced to the reliance on conspicuous low-level features through e.g. differences in contrast sensitivity or distinctive facial features. We have highlighted the importance of the inversion effects in the introduction (pp4-5, lines 104-110), and the presence of these inversion effects in the results (p15-16, lines 391-401) to make this point more clearly. 

The preliminary study where gender ratings were recorded did indeed involve greyscale faces in an oval aperture – this detail has been added on p5, line 123. 

These lapses relate to errors made by participants at large face sizes (where performance should otherwise be at ceiling). For psychometric function fitting, the lapse rate was 0 as standard. Some participants produced sufficient lapse errors that the resulting functions clearly diverged from the data points upon visual inspection. In these cases, the maximum allowable lapse rate was increased to either 0.05 or 0.1 in order to improve curve fitting. This was done on an individual basis, starting with 0.05 and increasing to 0.1 if the fits still did not improve. 7 participants had an adjusted lapse rate in Experiment 1, 3 in Experiment 2, and none in Experiment 3. Importantly, the adjusted lapse rate was applied across all conditions for that participant, meaning that neither the inversion effect nor the anisotropies in performance can be attributed to this factor. Clarification on this process has been added to the manuscript on page 7, lines 173-179.

We agree that investigating how face acuity thresholds vary over eccentricity would be interesting. In fact, we are currently conducting a series of experiments to examine this issue. The picture here is complicated by demonstrations that acuity is not the sole limitation on peripheral vision – rather, one must also consider crowding effects (particularly with complex stimuli such as faces) in order to understand our peripheral recognition abilities. Examining this in the current set of experiments would add a considerable number of conditions/experiments, significantly increasing the length and complexity of the manuscript. For clarity of the results and to ensure that polar angle variations were the main focus of the paper, we have chosen to explore eccentricity variations and their correlates in a subsequent study. Nonetheless, as the reviewer highlights, it is indeed the case that our (upright) face acuity thresholds are approximately 8-13 times larger than that of letter acuity at 10� eccentricity. We agree that this is an important link to make, and we now make mention of this relationship in the discussion on p17, lines 433-437. 

Finally, we have removed information about funding from the Acknowledgements section of the manuscript, and confirm that our funding statement should read: “This work was supported by the Biotechnology and Biological Sciences Research Council [grant number BB/J014567/1]”.

---

## [Decision Letter · Decision Letter 1]

24 Apr 2024

The resolution of face perception varies systematically across the visual field

PONE-D-23-36530R1

Dear Dr. Morsi,

We’re pleased to inform you that your manuscript has been judged scientifically suitable for publication and will be formally accepted for publication once it meets all outstanding technical requirements.

Kind regards,

PremNandhini Satgunam

Academic Editor

PLOS ONE

Additional Editor Comments (optional):

There is one minor correction required (see Reviewer note). Please do the needful.

Reviewers' comments:

Reviewer's Responses to Questions

**Comments to the Author**

1. If the authors have adequately addressed your comments raised in a previous round of review and you feel that this manuscript is now acceptable for publication, you may indicate that here to bypass the “Comments to the Author” section, enter your conflict of interest statement in the “Confidential to Editor” section, and submit your "Accept" recommendation.

Reviewer #1: All comments have been addressed

2. Is the manuscript technically sound, and do the data support the conclusions?

Reviewer #1: Yes

3. Has the statistical analysis been performed appropriately and rigorously? 

Reviewer #1: Yes

4. Have the authors made all data underlying the findings in their manuscript fully available?

Reviewer #1: (No Response)

5. Is the manuscript presented in an intelligible fashion and written in standard English?

Reviewer #1: Yes

6. Review Comments to the Author

Reviewer #1: The authors have addressed my comments.

There is a minor correction on

Line 459: [Alexia].

presumably a reference insert required

7. PLOS authors have the option to publish the peer review history of their article (what does this mean?). If published, this will include your full peer review and any attached files.

Reviewer #1: No

---

## [Editor Report · Acceptance letter]

30 Apr 2024

PONE-D-23-36530R1 

PLOS ONE

Dear Dr. Morsi, 

I'm pleased to inform you that your manuscript has been deemed suitable for publication in PLOS ONE. Congratulations! Your manuscript is now being handed over to our production team.

Kind regards, 

on behalf of

Dr. PremNandhini Satgunam 

Academic Editor

PLOS ONE